# Characterization of *Perionyx excavatus* Development and Its Head Regeneration

**DOI:** 10.3390/biology9090273

**Published:** 2020-09-05

**Authors:** Yun Seon Bae, Jung Kim, Jeesoo Yi, Soon Cheol Park, Hae-Youn Lee, Sung-Jin Cho

**Affiliations:** 1School of Biological Sciences, College of Natural Sciences, Chungbuk National University, Cheongju, Chungbuk 28644, Korea; bys@cbnu.ac.kr (Y.S.B.); yijeesoo32@chungbuk.ac.kr (J.Y.); 2Department of Molecular and Cell Biology, University of California, Berkeley, 142 Life Sciences Addition #3200, Berkeley, CA 94720-3200, USA; jkim81@berkeley.edu; 3Department of Life Science, Chung-Ang University, Seoul 06974, Korea; scpark@cau.ac.kr

**Keywords:** *Perionyx excavatus*, earthworm, embryonic development, head regeneration, juvenile

## Abstract

Regeneration is a biological process restoring lost or amputated body parts. The capability of regeneration varies among organisms and the regeneration of the central nervous system (CNS) is limited to specific animals, including the earthworm *Perionyx excavatus*. Thus, it is crucial to establish *P. excavatus* as a model system to investigate mechanisms of CNS regeneration. Here, we set up a culture system to sustain the life cycle of *P. excavatus* and characterize the development of *P. excavatus*, from embryo to juvenile, based on its morphology, myogenesis and neurogenesis. During development, embryos have EdU-positive proliferating cells throughout the whole body, whereas juveniles maintain proliferating cells exclusively in the head and tail regions, not in the trunk region. Interestingly, juveniles amputated at the trunk, which lacks proliferating cells, are able to regenerate the entire head. In this process, a group of cells, which are fully differentiated, reactivates cell proliferation. Our data suggest that *P. excavatus* is a model system to study CNS regeneration, which is dependent on the dedifferentiation of cells.

## 1. Introduction

Regeneration is the biological process of restoring damaged body parts. Organisms have different abilities for regeneration [1]. A few organisms restore the amputated region dramatically, whereas others heal the wound site without replacement. The cellular mechanisms of regeneration are categorized as morphallaxis (morphallactic regeneration) and epimorphosis (epimorphic regeneration) [2,3,4,5]. In morphallaxis, regeneration occurs mainly by the differentiation of pre-existing neoblasts. Conversely, epimorphosis is characterized by the dedifferentiation of adult tissue to form a highly proliferating cell mass called a blastema [4,5]. In some cases, the regenerative morphogenesis can occur in the combination of morphallaxis and epimorphosis [3,6].

*Hydra* and planarians completely regenerate a new organism from a small piece of a body, which explains their wide use in invertebrate regeneration research [7,8]. It is known that *Hydra* and planarians have numerous multipotent stem cells (interstitial cells or neoblasts) distributed all over the body for morphallactic regeneration. In contrast, vertebrate models, such as axolotl, newt, and teleost fish, show a limited capability of regeneration, which depends on the dedifferentiation of fate-determined adult cells [9,10,11].

*Perionyx excavatus* is a tropical earthworm with excellent regeneration capabilities [12,13]. More than 90% of *P. excavatus* regenerate and survive even after removal of the head part containing the central nervous system (CNS), which makes it a good material for the study of regeneration mechanisms. Here, we establish the culture system of earthworm, *P. excavatus* in the laboratory and sustain the life cycle of *P. excavatus*. We stage the embryogenesis of *P. excavatus* and characterize myogenesis, neurogenesis, and cell proliferation during development. Then, we investigate regeneration by performing an amputation of the animal’s trunk, which lacks proliferating cells. Interestingly, amputated *P. excavatus* fully regenerate the head part through the dedifferentiation of existing cells. We propose that *P. excavatus* is a novel invertebrate model organism that regenerates through epimorphosis.

## 2. Materials and Methods

### 2.1. Cultivation of P. excavatus

Adults specimens of *P. excavatus*, which were collected in The Nanji Water Reclamation Center (Seoul, Korea), were bred in the Laboratory of Cellular and Developmental Biology (LCDB) at Chungbuk National University. Adults were raised in plastic containers of 60 × 40 × 15 cm^2^ filled with organic soil of at least 70% humidity, which was provided from The Nanji Water Reclamation Center. Organic soil was changed every 2 weeks. The incubation room was kept at 22–26 °C.

### 2.2. Embryo and Juvenile Preparation

Only mature individuals with well-developed clitellum were moved into a box that was filled with the organic matter and peat moss. The top of the box was covered with non-woven fabric to maintain a dark environment. Cocoons were deposited for 14 days and hatched embryos were recovered at favorable stages. For juvenile experiments, slightly pigmented earthworms, which were aged for 7–10 days after hatching, were collected.

### 2.3. Regeneration of Juvenile Head

With a clean razor blade, the trunk region, which is 10 segments after the last pumping vessel, was cut. After the extirpation of the head part, the remaining body part was incubated in a 90-mm plastic petri-dish (SPL Life Science, 10095, Pocheon, Gyeonggi-do, Korea) with water-soaked filter paper (Advantec, 00021110, Chiyoda, Tokyo, Japan). Filter papers were changed once a day and petri-dishes were kept at 22–25 °C.

### 2.4. Sample Preparation for Imaging

Embryos and juveniles were fixed in 4% PFA in 1× PBS (140 mM NaCl, 6.5 mM KCl, 2.5 mM Na_2_HPO_4_, 1.5 mM KH_2_PO_4_, pH 7.5) for 50 min at room temperature, washed 3 times with PBT (PBS + 0.1% TritonX-100) for 20 min each and used for staining or stored in PBT at 4 °C for up to 7 days. Juveniles were soaked for 50 min in a relaxant buffer (4.8 mM NaCl + 1.2 mM KCl + 10 mM MgCl_2_ + 8% EtOH) for clear visualization before 4% PFA fixation.

### 2.5. EdU Assay for Cell Proliferation

For EdU incorporation, embryos and juveniles were incubated in the EdU solution with adjustment of media concentration and incubation time according to the animal size (embryos: 50 mM of EdU for 48 h, 1 day post amputation (dpa) juveniles: 150 mM of EdU for 24 h, 3–14 dpa juveniles: 200 mM of EdU for 48 h). Samples were fixed in 4% PFA for 2.5 h. After 3 cycles of washing in PBT (1× PBS + 0.1% TritonX-100), samples were incubated in the solution containing 5% β-mercaptoethanol and 1% TritonX-100 in 0.1 M Tris-HCl (pH 7.5) at 37 °C for 1.5 h on a shaker (70 rpm) to soften the chitin of the cuticle. After washing, the labeling reaction was performed according to the manufacturer’s protocol (Click-iT^®^ EdU Imaging Kit, Thermo Fisher Scientific, #C10339, Waltham, MA, USA). Then, samples were washed withs PBT and counterstained using Hoechst (1.5:2000) for labeling nuclei.

### 2.6. Phalloidin Staining and Immunohistochemistry

To detect F-actin, we incubated fixed embryos in Texas-red-conjugated phalloidin (1:100; Sigma, T7471, St. Louis, MO, USA) overnight. For immunohistochemistry, fixed samples were incubated in the blocking solution (Sigma, 11096176001, St. Louis, MO, USA) overnight, then incubated with mouse anti-acetylated α-tubulin antibody (1:800 in blocking solution; Sigma, T-7451, St. Louis, MO, USA) or rabbit anti-FMRF-amide (1:2000 in blocking solution; Sigma, AB15348, St. Louis, MO, USA) at room temperature for 2–4 days. Samples were washed 3 times for 10 min and overnight in PTA (1× PBS + 0.1% TritonX-100 + 0.1% NaN₃). After several washes, the secondary antibody (1:200 in PBT; anti-mouse Alexa 488 for green and 1:500 in PBT; anti-rabbit Alexa 568 for red) was applied for 2 days at room temperature. For nuclei staining, samples were incubated in DAPI solution (1:1000 in PBT; Thermo Fisher Scientific, D1306, Waltham, MA, USA) for 30 min. After several times of wash in PBT, samples were mounted in Fluoromount-G (Southern Biotech, Birmingham, AL, USA).

### 2.7. Microscopy

All images (brightfield and fluorescence) were taken on a Nikon SMZ18 microscope (Shinagawa, Tokyo, Japan), Leica DM6 B microscope (Wetzler, Hesse, Germany), and Invitrogen Evos FL Auto2 microscope (Waltham, MA, USA).

## 3. Results and Discussions

### 3.1. The Life Cycle of Perionyx excavatus

First, we characterized the life cycle of *P. excavatus* in the laboratory environment (Figure 1).

Sexually matured clitellate worms were maintained in the organic soil for mating and deposited cocoons 28 days after mating. Cocoons were transferred to a Petri-dish where they underwent embryonic development for 14 days to become juveniles. One or two juveniles hatched from each cocoon and the viability of embryos in cocoons was about 70%. Juveniles were transferred back to the organic soil for further maturation. Thirty days later, adult earthworms (with clitellum) were observed. Our data were consistent with the previously reported life cycle of *P. excavatus* [14].

### 3.2. Characterization of Embryonic Development of Perionyx excavatus

Next, we investigated and staged the embryonic development of *P. excavatus* based on morphological characteristics (Figure 2). In the following figures, animals are shown with the anterior end at the left and the dorsal side at the top.

The gastrulation stage exhibited blastopore formation (gastrulation (GA)). The anterior of blastopore remained open for the mouth formation (E1). Stage 2 embryos (E2) showed the elongated germ band along the ventral midline after closing of germ layers. Stage 3 embryos (E3) were slightly larger than E2 embryos, formed the stomodaeum (or pharynx) and segmentation started on the epidermal surface. Stage 4 and 5 embryos (E4, E5) showed precise segmentation, an elongated mouth part, and acquired a kidney shape. During the embryonic stage 5 (E5), the segmented area of the germinal plate was increased. Stage 6 embryos (E6) exhibited a wormlike shape in the anterior part, while the dorsal side of the midpart was bent upwards. Subsequently, segment formation and embryo elongation proceeded from anterior to posterior in stage 7–9 embryos (E7–E9). All organs, such as pumping vessels, dorsal vessel, and the intestine, were developed and pigmentation started at the juvenile stage (J). Finally, the segmented juvenile worms hatched at around day 14 (J).

To establish a model system for regeneration, it is necessary to compare the embryonic development and post-embryonic development during the regeneration process, which occurs in adulthood. By observing the morphogenesis of embryos, although the cleavage of the zygote cannot be observed, we characterized the developmental process of *P. excavatus*, which was similar to that of other Annelida [15,16,17,18,19,20,21].

### 3.3. Embryonic Myogenesis and Neurogenesis of Perionyx excavatus

To characterize myogenesis in *P. excavatus* embryos, we performed phalloidin staining (Figure 3a). First, the network of primary muscles essential for nutrient uptake in a cocoon was apparent in E2 [15]. Next, the muscles arising from the germinal bands were observed in the ventral side of E4. Further primary longitudinal muscle fibers were scattered over the trunk of embryos in E5. Prominent ventral muscle strands were connected to the circular mouth muscle (buccal muscle) in this stage, which is a characteristic in Annelida [15,18]. The segmented structure of the circular and longitudinal muscle fibers was seen along almost the entire anteroposterior axis in E5 and E6. The number and density of circular and longitudinal muscles increased as the germinal plate closed in E6 and E7. This process reflects the general mode of development in *Eisenia andrei* and *Enchytraeus coronatus*; the segmental development of the circular muscles from initial muscles at the segment borders, as well as a continuous development of primary longitudinal muscles from anterior pole [15,18]. The organization (regularly arranged fine circular and longitudinal muscles) was first established ventrally from the germinal plate for circular, as well as longitudinal muscle strands [15,18].

Then, we investigated the neurogenesis of *P. excavatus* embryos by immunohistochemical analysis of acetylated α-tubulin, which is found in nephridia, external sensory hairs, nerves of the cerebral ganglion and peripheral nervous system (Figure 3b). In embryogenesis, the anlagen of the brain (supraesophageal ganglion) appeared and the two prominent ventral nerve cords were observed in the anterior part of E2. The brain and ventral nerve cord get connected by circumesophageal connectives and the anlagen of nephridia were formed from anterior to posterior in E3. The development of segmental nephridia continued in E3–E9. In E5–E7, the brain and ventral nerve cord became distinct and the palp nerves extended from the circumesophageal connectives. In E9, the formation of the CNS with nephridia was complete. Our results showed the embryonic development of *P. excavatus* was similar to that of other earthworms, such as *Eisenia andrei* [15] or *Enchytraeus japonensis* [17].

### 3.4. Cell Proliferation in Embryos and Juveniles of Perionyx excavatus

The EdU staining, which visualizes cell proliferation, revealed specific labeling patterns corresponding to some developing structures and organs (Figure 4a). In neurula stages (after gastrulation stage), the ventral organs, including ventral ganglion cells, nephridia and the hypocerebral organs, showed a high number of EdU-positive cells. However, in the trunk, the number of EdU-positive cells was greatly reduced and these proliferating cells were observed only in the head and tail part, not in the trunk (Figure 4b,c) [22]. This result suggests that organs located in the anterior and posterior grow actively. The pattern of EdU-positive cells in *P. excavatus* juveniles is different from other regeneration models [23]. In *Hydra* and planarians, which maintain neoblast cells throughout the body, the regeneration is initiated from these pre-existing neoblasts [2].

### 3.5. Cell Proliferation in Regenerative Tissues of Perionyx excavatus

To address whether the regeneration of earthworms is dependent on neoblasts (morphallaxis) or dedifferentiation of existing cells (epimorphosis), EdU labeling was performed in regenerative tissues from the amputated anterior part. Juveniles of *P. excavatus* were amputated between the 10th and 11th segments after pumping vessels at the trunk region, where cell proliferation was not detectable (Figure 5a). At 1 day post-amputation (dpa), cell proliferation was observed only at the edge of regenerating tissue in the epidermis covering the wound site, whereas the undamaged trunk region remained EdU-negative (Figure 5b). This finding suggests that existing cells undergo dedifferentiation in order to regenerate. After wound healing, the EdU-positive cells were most abundant at the ventrolateral epidermis in the blastema-forming area at 3 dpa. Then, the EdU-positive cells concentrated at the prostomium and mouth at 5 dpa and active cell proliferation was maintained to form organs until 14 dpa. Our results suggest that *P. excavatus* undergoes epimorphosis after anterior amputation, which is similar to other invertebrate models of regeneration [24,25,26,27].

### 3.6. Head Regeneration of Juveniles of Perionyx excavatus

To investigate the reconstruction of the neural network during regeneration, we performed immunohistochemistry using antibodies against acetylated α-tubulin and FMRF-amide (Phe-Met-Arg-Phe), which have high specificities for the nervous system (Figure 6a,b).

The CNS of *P. excavatus* is composed of a cerebral ganglion (brain) located dorsally and a ganglionic ventral nervous system reaching up to the pygidium, which is connected via circumesophageal connectives. Palp nerves originate from the neuropil of the circumesophageal connectives and extend to the whole prostomium [28]. During anterior regeneration, nerve fibers began to extend from the remaining ventral nerve cord at 1 dpa. During blastema stage at 3 dpa, nerve fibers formed loops connecting circumesophageal connectives of the ventral nerve cord and the neural network covered the entire blastema [17]. At 5 dpa, the circumesophageal connectives were fused dorsally and the first cerebral commissure (the primordium of the prospective brain) appeared before the segmentation of blastema [17,28,29,30]. At 7 dpa, the regenerated nervous system was stretched to the anterior direction and the palp nerves were visible. As the anterior regeneration proceeded, the regenerated nervous system was extended with further segmentation. At 10 dpa, the segmental ganglia were highly visible in the regenerated segments. Both circumesophageal connectives and the brain were apparent, with the formation of 13 segments in the anterior head part and no more segments were added. The anterior nervous system was fully regenerated at 14 dpa.

There are some differences between embryogenesis and regeneration in the development of the CNS. In embryogenesis, the anlagen of the brain and ventral nerve cord appear first, followed by the development of circumesophageal connectives (Figure 6), whereas in head regeneration, the circumesophageal connectives and ventral nerve cord develop first, with the distinct brain after segmentation [30,31,32,33]. In addition, Myohara *et al*. reported that segmentation occurs in sequence from anterior to posterior in embryogenesis but simultaneously for the seven head segments of E. japonensis in regeneration [17]. We also observed the simultaneous segmentation of the 13 head segments in regeneration, during anterior to posterior segmentation in embryogenesis.

## 4. Conclusions

We firstly characterized and staged embryonic development, including myogenesis and neurogenesis, from gastrulation to the juvenile stage in *P. excavatus* using phalloidin staining, immunohistochemistry, and the EdU assay. In embryogenesis, cell proliferation occurs actively in developing organs, such as the CNS, segmental nephridia, and mesodermal and ectodermal cells, whereas juveniles maintain proliferating cells in the head and tail parts, not in the trunk. We observed that the CNS develops differently in embryogenesis and head regeneration. In embryogenesis, the anlagen of the brain and ventral nerve cord appear first, followed by the development of circumesophageal connectives, whereas the circumesophageal connectives and ventral nerve cord develop simultaneously during the head regeneration. Epimorphosis is accompanied by the mitotic activity of cells and the formation of a highly proliferating cell mass called blastema, while morphallaxis is accompanied by the rearrangement of cells in the original body fragments without blastema formation [5,6,34]. In this work, our data suggest that *P. excavatus* exhibits epimorphosis during head regeneration because the dedifferentiation of mature cells is observed in the amputated trunk. Further, *P. excavatus* undergoes complete regeneration, including reconstruction of important organs (i.e., CNS, nephridia, pumping vessels), within 2 weeks of amputation. This study provides an essential basis for understanding the regeneration mechanism and suggests an important model system for regenerative studies.

## Figures and Tables

**Figure 1 biology-09-00273-f001:**
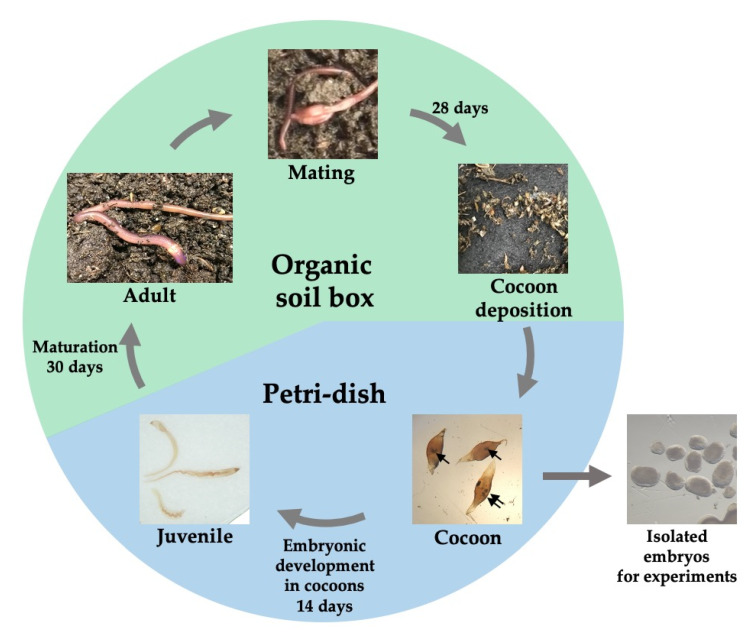
The life cycle of *P. excavatus.* Arrows indicate embryos in cocoons.

**Figure 2 biology-09-00273-f002:**
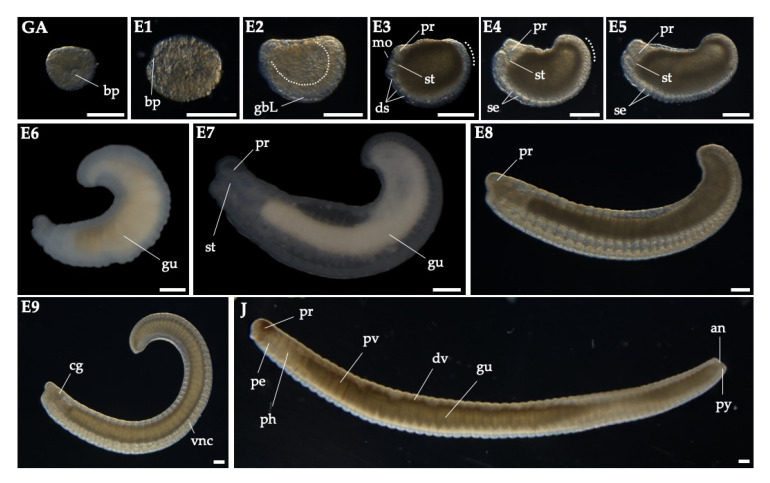
The embryonic development of *P. excavatus* was staged based on its morphology. Gastrulation (GA), the embryonic stage 1–9 (E1–9), and the juvenile stage (J). Gastrulation stage is shown in ventral view. Embryos are shown with the anterior end at the left and the dorsal side at the top. Abbreviations: an, anus; bp, blastopore; cg, cerebral ganglion; ds, dissepiment; dv, dorsal vessel; gbL, left germinal band; gu, gut; mo, mouth; pe, peristomium; ph, pharynx; pr, prostomium; pv, pumping vessel; py, pygidium; se, segment; st, stomodaeum. Scale bar, 100 μm.

**Figure 3 biology-09-00273-f003:**
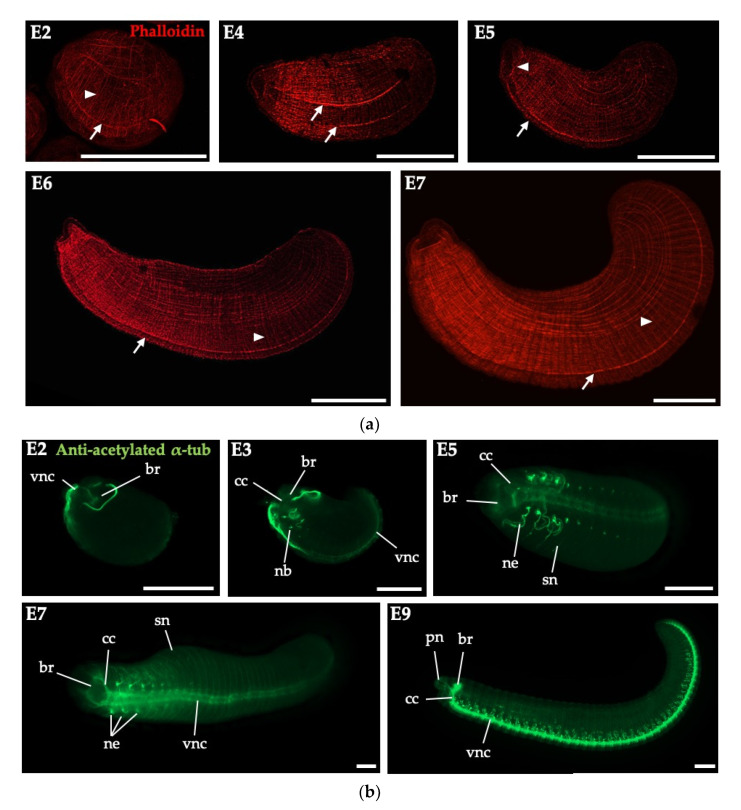
Myogenesis (**a**) and neurogenesis (**b**) of *P. excavatus*. (**a**) Myogenesis was visualized by phalloidin staining (red) in embryonic stages of *P. excavatus*. Arrows and arrowheads indicate longitudinal muscles (along the anterior–posterior axis) and circular muscles (perpendicular to the anterior–posterior axis), respectively. Scale bar, 200 μm. (**b**) Nervous system development of *P. excavatus* visualized by immunohistochemistry using anti-acetylated α-tubulin (green). Abbreviations: br, brain; cc, circumesophageal connective; nb, nephroblast; ne, nephridium; pn, palp nerves; sn, segmental nerves; vnc, ventral nerve cord. Scale bar, 200 μm.

**Figure 4 biology-09-00273-f004:**
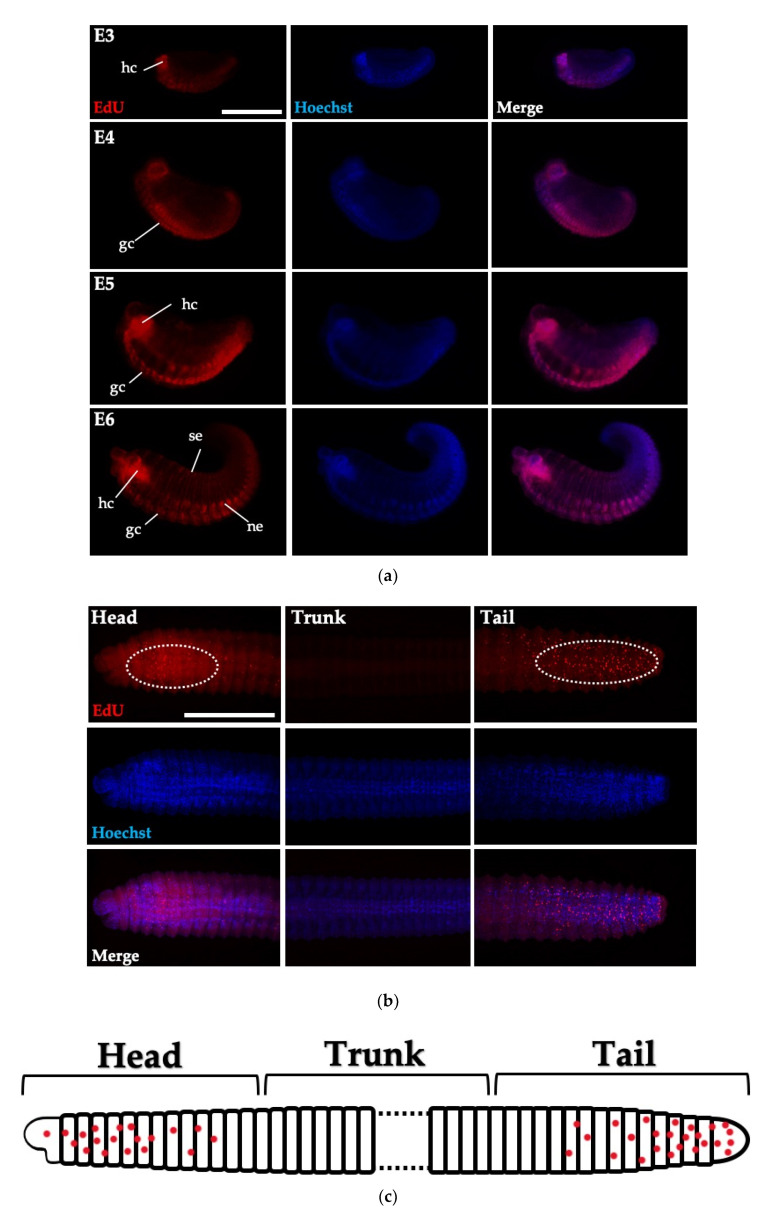
The pattern of cell proliferation in embryo and juvenile of *P. excavatus.* (**a**) Proliferating cells are visualized using the EdU assay (red) and Hoechst stained DNA (blue) during embryogenesis. Abbreviations: gc, ganglionic cell; hc, hypocerebral ganglionic cell; ne, nephridium; se, segment. Scale bar, 200 μm. (**b**) Cell proliferation in the head, trunk, or tail of juvenile of *P. excavatus* is visualized using EdU assay. Scale bar, 500 μm. (**c**) The schematic cartoon shows red spots representing cell proliferation in the juvenile. The head and tail part of the juvenile shows EdU-positive proliferating cells, whereas EdU-positive cells in the trunk are not detectable (**b**,**c**). Scale bar, 200 μm.

**Figure 5 biology-09-00273-f005:**
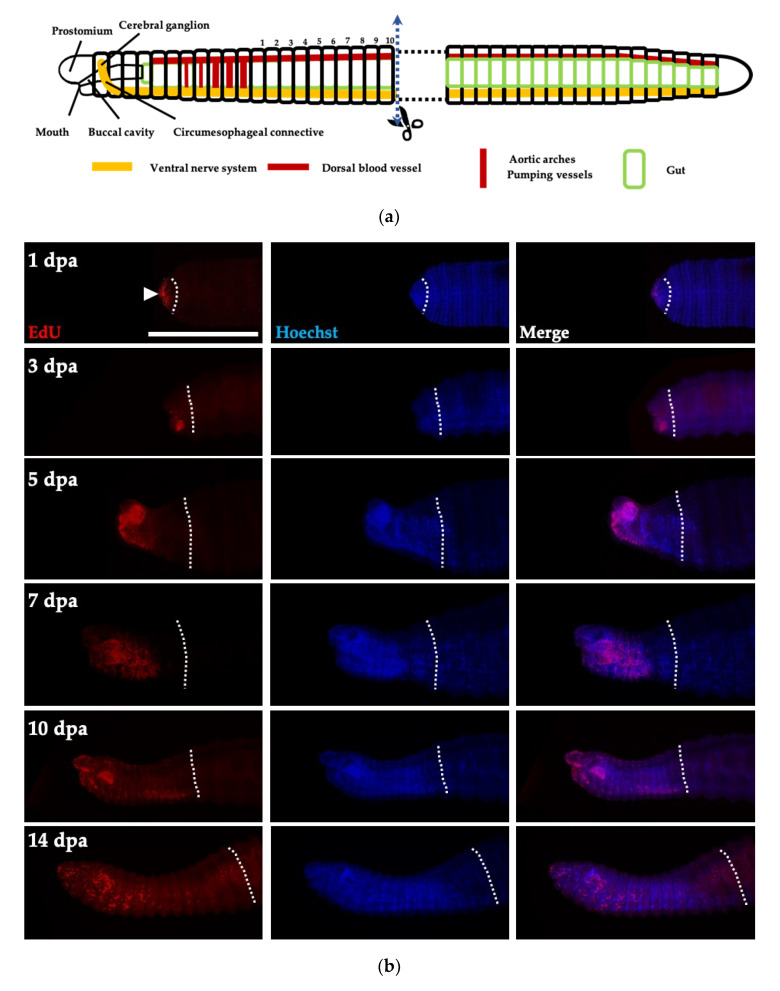
Active cell proliferation in regeneration. (**a**) The schematic cartoon indicating the amputation site, which is located in the trunk region, lacking cell proliferation. (**b**) Cell proliferation in regenerating tissues is visualized by EdU staining (red) at indicated days post-amputation (dpa). Nuclei are marked by Hoechst staining (blue). Dotted lines indicate the site of amputation. Scale bar, 500 μm.

**Figure 6 biology-09-00273-f006:**
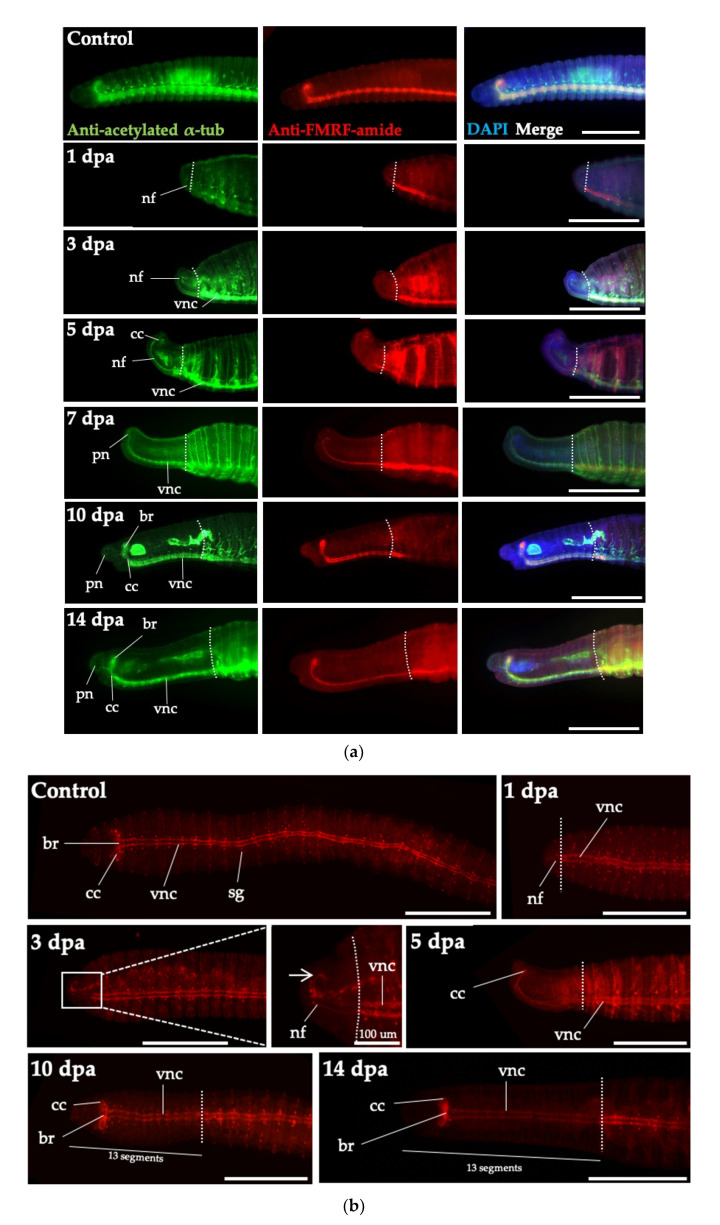
Visualization of nervous system regeneration. (**a**) The nervous system regeneration in the head region of *P. excavatus* was visualized using anti-acetylated *α*-tubulin staining (green) and anti-FMRF-amide staining (red) at indicated dpa. Nuclei are marked by DAPI staining (blue). Dotted lines represent the position of amputation. At 1 dpa, the wound at the amputated site is healed. The nervous system regenerates and nerve connections from the brain to the ventral nerve cord are clearly visualized. Regenerative tissues are shown in lateral view, with the dorsal side on the top. Scale bar, 500 μm. (**b**) The process of nervous system regeneration in the head region of *P. excavatus* was visualized using anti FMRF-amide staining (red). At 3 dpa, the white arrow indicates the loop of circumesophageal connectives. Regenerative tissues are shown in ventral view. Abbreviations: br, brain; cc, circumesophageal connective; nf, nerve fibers; pn, palp nerves; sg, segmental ganglia; vnc, ventral nerve cord. Scale bar, 500 μm.

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
