# Peer review of "Characterization of Perionyx excavatus Development and Its Head Regeneration"

_biology, 2020, doi:10.3390/biology9090273_

Round 1

Reviewer 1 Report

The article "Characterization of Perionyx excavatus Development and its Head Regeneration" by Yun Seon Bae, Jung Kim, Jee Soo Lee, Soon Cheol Park, Hae-Youn Lee, Sung-Jin Cho details neuro- and myogenesis of the oligochaete Perionyx excavatus as well as highlights the species' remarkable ability to regenerate the anterior part of the body including the nervous system. In contrast to planarians, these earth worms dedifferentiate, migrate and redifferentiate the cells in these regions, which is a pretty amazing find in itself. This is a great study, which adds quite a lot of information to our current knowledge.

This study is well-thought through and well-done with appropriate set of experiments. The study furthermore lists a lot of relevant literature, though fails to discuss the own results with published data in more detail. I would recommend to describe some of the results (myogenesis, neurogenesis - neuropile architecture) in more detail, adding information about where tracts extend to or what they possibly connect to, as well as to discuss them with the published results more specifically - e.g. how the regeneration of the central nervous system related to the embryonic development and how this pattern is followed by other species (oligo- and polychaetes). 

In general this is a great study and an important contribution to annelid neurobiology, regeneration biology and cell differentiation. I therefore recommend the manuscript for publication given that few things (I also added some comments in the attached PDF) are taken care of. 

Author Response

Editor in Chief

MDPI_biology

Dear Editor,

RE: Mauscript ID: biology-905543

Thanks very much for your letter on August 11, 2020.

Our manuscript (Mauscript ID: biology-905543) entitled “Characterization of Perionyx excavatus Development and its Head RegenerationMuscular development in Urechis unicinctus (Echiura, Annelida)” has been revised for submission in MDPI_biology. The reviewer’s comments were valuable in assisting me in the revision. The followings are changes that have been made in response to reviewers' comments.

- Above all, we appreciate your valuable advice.

- The following is our point-to-point response to the reviewers.

- In all the revised manuscript, we have changed sentences that have been highlighted.

- All the issues were addressed following the reviewer suggestions.

We tried our best to improve the manuscript and made some changes to the manuscript. These changes will not influence the content and framework of the paper. We appreciate for Reviewers’ warm work earnestly and hope all corrections and revisions in the revised manuscript will be satisfactory to biology. Once again, thank you very much for your comments and suggestions.

Sincerely yours,

Sung-Jin Cho, Ph.D.

Associate Professor,

Department of Biology(S1-5,204b),

College of Natural Sciences,

Chungbuk National University,

52 Naesudong-ro, Heungdeok-gu,

Cheongju, Chungbuk 361-763,

Republic of Korea

E-mail:sjchobio@chungbuk.ac.kr

Reviewer #1

The article "Characterization of Perionyx excavatus Development and its Head Regeneration" by Yun Seon Bae, Jung Kim, Jee Soo Lee, Soon Cheol Park, Hae-Youn Lee, Sung-Jin Cho details neuro- and myogenesis of the oligochaete Perionyx excavatus as well as highlights the species' remarkable ability to regenerate the anterior part of the body including the nervous system. In contrast to planarians, these earth worms dedifferentiate, migrate and redifferentiate the cells in these regions, which is a pretty amazing find in itself. This is a great study, which adds quite a lot of information to our current knowledge.

This study is well-thought through and well-done with appropriate set of experiments. The study furthermore lists a lot of relevant literature, though fails to discuss the own results with published data in more detail. I would recommend to describe some of the results (myogenesis, neurogenesis - neuropile architecture) in more detail, adding information about where tracts extend to or what they possibly connect to, as well as to discuss them with the published results more specifically - e.g. how the regeneration of the central nervous system related to the embryonic development and how this pattern is followed by other species (oligo- and polychaetes). 

In general this is a great study and an important contribution to annelid neurobiology, regeneration biology and cell differentiation. I therefore recommend the manuscript for publication given that few things (I also added some comments in the attached PDF) are taken care of. 

Response: First of all, we are grateful that the reviewer finds this work interesting. We agree with the reviewer’s constructive comments, which recommend describing and discussing our results in more detail by comparing the published literature. We have revised the manuscript according to all suggestions, and include a detailed point-by-point response to all comments below. In addition, we carefully rechecked the manuscript for any syntax errors in English.

Detail response of peer-review-8278487.v1 PDF file.

Line 76

Mistyping in Materials and Methods

Response: The mistyping was modified as suggested.

Line 85

does this stand for "day past amputation"?

Response: This abbreviation means “day post amputation” and the term has been stated.

Line 99, 100

does this mean you ended up with dilutions of 1:4000 and 1:10000, respectively, as final concentrations?

Response: It has been modified to avoid confusion.

Line 127

Misspelled

Response: It has been modified

Line 131

The species name should stand out from the normal text, thus not be italic in this case

Response: The scientific name has been modified as suggested.

Figure 3b

line to the structure is missing

Response: It has been modified.

Line 148-150

it would help with orientation if a few muscles are labelled and not only indicated by arrowheads

Response: The additional description has been added as follows:

“Arrows and arrowheads indicate longitudinal muscles (along the anterior-posterior axis) and circular muscles (perpendicular to the anterior-posterior axis), respectively.

Line 152, 153

maybe arrange these alphabetically

Response: It is arranged alphabetically as suggested.                    

Line 170, 171, 172

this is super-interesting: In some of the studies poychaetes, the first serotonergic elements are found around the first commissure (usually in the first few segments, such as in Owenia, Dinophilus or Phascolion) - here it looks much more posterior. Could you possibly detail the locallisation of the onset of ventral nerve. cord formation a bit more? And since you also used FMRFamide, could you detect the location of the first FMRFamidergic neurons (somata and neurites)?

Response: We appreciate for your important comments and correction in this data.

Compared to other pictures of same stages in our data, as you referred, we also observed the analgen of brain and ventral nerve cord in the anterior part of E2. Thus, the E2 stage of Fig.3b was replaced with another picture for localization of ventral nerve. And we have changed the sentence as follows:

“In embryogenesis, the anlagen of the brain (supraesophageal ganglion) appeared, and the two prominent ventral nerve cords were observed in the anterior part of E2.”

We didn’t observe the FMRF amide expression pattern in nervous system. We hope to investigate detailed neurogenesis using specific antibodies in our further study.

We are sorry for the confusion caused by the figure 3b in our manuscript.

Line 172, 173

Misspelled word

Response: The words have been modified as suggested.

Line 174

Segmental and deleted the words

Response: The sentence has been modified as suggested (segmental nephridia)

Line 190, 164, 165

Misspelled

Response: The sentence has been modified

Line 191-194

This sentence is not clearly phrased - I am not sure whether some words are missing or whether it is just hard to follow.

Response: To clarify, the sentence was rephrased as follows:

“However, the number of EdU-positive cells was greatly reduced, and these proliferating cells were observed only in the head and tail part, not in the trunk (Fig. 4b,c) [22]. This result suggests that organs located in the anterior and posterior grow actively”

Line 195

since this refers to a species, this should be spelled with capital H and in italics

Response: spelled with capital H and P in italics

Figure 4

maybe consider using another colour scheme for representing EdU here - the small/weak red dots are hard to see, especially in the merged image - possibly white or glow will give better results

Response: The figure has been modified as suggested by increasing the brightness.

Line 181, 172, 174, 175,187

Uncorrected Figure legend

Response: The figure legend has been modified as suggested.

Line 183

consider arranging abbreviations alphabetically

Response: It has been arranged alphabetically as suggested.                       

Line 198

As above - should be non-italic in this case

Resopnse: It has been modified

Line 204

Misspelled

Response: It has been modified

Line 211, 212

Most abundant

Response: It has been modified

Line 199, 200

Uncorrected Figure legend

Response: The figure legend has been modified as suggested.

Line 217

needs to be non-italic

Response: has been modified

Line 222, 224

Misspelled and wrong preposition

Response: Misspelled words and wrong preposition has been modified

Line 228

since tubullin is a marker for the cytoskeleton and thus reveals mainly structures with surface extensions (cilia, ...), this is not that surprising. In contrast to the neuropeptide FMRFamide, which really is limited to (parts of) the nervous system, tubulin can thereby be found in a lor of tissues (gland cells, ciliated cells in the epidemris, digestive system, nephridia, ...

Response: The sentence has been deleted.

Line 229, 230, 231

maybe arrange abbreviations alphabetically

Response: The abbreviations has been arranged as mentioned.

Line 231

no image-markers are used, but I guess you mean the one stating 100µm above the scale bar

Response: has been deleted

Line 237

circumesophageal

Response: has been modified

Line 251, 253

Should be Italic

Response: has been modified

Line 260

Should be modified

Response: The words have been modified as suggested.

Line 259, 260, 261

please rephrase - it is hard to understand what you mean

Response: As mentioned in the response on Line 238, we add the sentence to clarify the explanation about “in the head and tail parts in the juveniles” as follows:

“In embryogenesis, cell proliferation occurs actively in developing organs, such as the CNS, segmental nephridia, and mesodermal and ectodermal cells, whereas juveniles maintain proliferating cells in the head and tail parts, not in the trunk”

Page 13

Reference misspelled

Response: Every misspelled reference has been modified as suggested.

Reviewer 2 Report

The manuscript contains valuable experimental results but their interpretation is not the best.

Timing of the embryonic development depends on the differentiation and closing of germinal layers and it can be followed by the observation of the segmentation as was described by Prosser CL (1933) and supported by some recent studies (Boros et al. 2008, 2010).

The regeneration of the central nervous system is coupled with the segment renewing in earthworms as it was established by Chapron (1971).

Some sentences are not clear e.g. 218 and 209 rows: At 5 dpa the brain was regenerated at the anterior edge of the circumesophageal connectives, and the redifferentiated prostomium was observed. The primordium of the prospective brain appears at the dorsal part of the regeneration blastema independently from the ventral nerve cord. Please overview Chaprons’ description!

It would be useful if the authors used the concepts correctly like epimorphosis and morphallaxis. Epimorphosis is characterized by extensive cell proliferation (Sunderland ME (2010). "Regeneration: Thomas Hunt Morgan's window into development". Journal of the History of Biology. 43 (2): 325–61. doi:10.1007/s10739-009-9203-2. PMID 20665231.) so the next sentence is not clear (rows 238 and 239): In head regeneration, P. excavatus can regenerate by epimorphosis involving dedifferentiation, blastema formation, and redifferentiation.

A more comprehensive rewriting of the manuscript is absolutely needed.

There are some important studies (see below) that can be used during the rewriting of the manuscript.

Chapron, C., 1971. Etude de l'origine et de la différentiation du pharynx au cours de la régénération chez l'annélide Eisenia fœtida. Development, 25(3), pp.439-455.

Martinez, V. G., Menger III, G. J., & Zoran, M. J. (2005). Regeneration and asexual reproduction share common molecular changes: upregulation of a neural glycoepitope during morphallaxis in Lumbriculus. Mechanisms of Development, 122(5), 721-732.

Martinez, V. G., Manson, J. M., & Zoran, M. J. (2008). Effects of nerve injury and segmental regeneration on the cellular correlates of neural morphallaxis. Journal of Experimental Zoology Part B: Molecular and Developmental Evolution, 310(6), 520-533.

Kostyuchenko, R. P., & Kozin, V. V. (2020). Morphallaxis versus epimorphosis? Cellular and molecular aspects of regeneration and asexual reproduction in annelids. Biology Bulletin, 47(3), 237-246.

Boros, A., Reglodi, D., Herbert, Z., Kiszler, G., Nemeth, J., Lubics, A., Kiss, P., Tamas, A., Shioda, S., Matsuda, K. Pollak, E., and Molnar L. (2008). Changes in the expression of PACAP-like compounds during the embryonic development of the earthworm Eisenia fetida. Journal of molecular neuroscience, 36(1-3), 157-165.

Boros, Á., Somogyi, I., Engelmann, P., Lubics, A., Reglodi, D., Pollák, E., & Molnár, L. (2010). Pituitary adenylate cyclase-activating polypeptide type 1 (PAC1) receptor is expressed during embryonic development of the earthworm. Cell and tissue research, 339(3), 649-653.

Prosser CL (1933) Correlation between development of behavior and neuromuscular differentiation in embryos of Eisenia foetida, Sav. J Comp Neurol 58:603–641

Author Response

Editor in Chief

MDPI_biology

Dear Editor,

RE: Mauscript ID: biology-905543

Thanks very much for your letter on August 11, 2020.

Our manuscript (Mauscript ID: biology-905543) entitled “Characterization of Perionyx excavatus Development and its Head RegenerationMuscular development in Urechis unicinctus (Echiura, Annelida)” has been revised for submission in MDPI_biology. The reviewer’s comments were valuable in assisting me in the revision. The followings are changes that have been made in response to reviewers' comments.

- Above all, we appreciate your valuable advice.

- The following is our point-to-point response to the reviewers.

- In all the revised manuscript, we have changed sentences that have been highlighted.

- All the issues were addressed following the reviewer suggestions.

We tried our best to improve the manuscript and made some changes to the manuscript. These changes will not influence the content and framework of the paper. We appreciate for Reviewers’ warm work earnestly and hope all corrections and revisions in the revised manuscript will be satisfactory to biology. Once again, thank you very much for your comments and suggestions.

Sincerely yours,

Sung-Jin Cho, Ph.D.

Associate Professor,

Department of Biology(S1-5,204b),

College of Natural Sciences,

Chungbuk National University,

52 Naesudong-ro, Heungdeok-gu,

Cheongju, Chungbuk 361-763,

Republic of Korea

E-mail:sjchobio@chungbuk.ac.kr

Reviewer #2

The manuscript contains valuable experimental results but their interpretation is not the best.

Response: We thank the reviewer for his/her time and effort in reviewing our paper and the constructive comments.

Timing of the embryonic development depends on the differentiation and closing of germinal layers and it can be followed by the observation of the segmentation as was described by Prosser CL (1933) and supported by some recent studies (Boros et al. 2008, 2010).

Response: We agree. Based on articles you sent, paragraph has been revised by adding more detailed explanations about the germ layer closure and segmentation. And the references have been added as suggested.

“The gastrulation stage exhibited blastopore formation (GA). The anterior of blastopore remained open for the mouth formation (E1). Stage 2 (E2) embryos showed the elongated germ band along the ventral midline after closing of germ layers. Stage 3 embryos (E3) were slightly larger than E2 embryos, formed the stomodaeum(or pharynx), and segmentation started on the epidermal surface”

The regeneration of the central nervous system is coupled with the segment renewing in earthworms as it was established by Chapron (1971).

Response: We apologize that we cannot include the Chapron (1971) paper, because it is written in French, which is the language that we cannot read. Instead, we added these alternative papers (Myohara, 2004; Muller, 2004, Meyer, 2009, Weidhase, 2015;) to support the idea that the CNS regeneration is accompanied by renewing segments.

The paragraph has been modified as follows:

“At 5 dpa, the circumesophageal connectives were fused dorsally and the first cerebral commissure (the primordium of the prospective brain) appeared before the segmentation of blastema [17, 28, 29, 32]. At 7 dpa, the regenerated nervous system was stretched to the anterior direction, and the nerve fibers in the palps were visible. As the anterior regeneration proceeded, the regenerated nervous system was extended with further segmentation.”

Some sentences are not clear e.g. 218 and 219 rows: At 5 dpa the brain was regenerated at the anterior edge of the circumesophageal connectives, and the redifferentiated prostomium was observed. The primordium of the prospective brain appears at the dorsal part of the regeneration blastema independently from the ventral nerve cord. Please overview Chaprons’ description!

Response: Since we cannot understand Chapron’s paper written in French, we carefully compared our data to alternative papers: Myohara, 2004 and Muller, 2004.

For clarification, the sentence has been modified as follows:

“At 5 dpa, the circumesophageal connectives were fused dorsally and the first cerebral commissure (the primordium of the prospective brain) appeared before the segmentation of blastema [17, 28, 29, 32].

It would be useful if the authors used the concepts correctly like epimorphosis and morphallaxis. Epimorphosis is characterized by extensive cell proliferation (Sunderland ME (2010). "Regeneration: Thomas Hunt Morgan's window into development". Journal of the History of Biology. 43 (2): 325–61. doi:10.1007/s10739-009-9203-2. PMID 20665231.) so the next sentence is not clear (rows 238 and 239): In head regeneration, P. excavatus can regenerate by epimorphosis involving dedifferentiation, blastema formation, and redifferentiation.

Response: We agree that the major characteristic of epimorphosis is the formation of a blastema, a mass of proliferating cells. In the recent review paper, the blastema formation is accomplished by dedifferentiation and redifferentiation of somatic cells (Ribeiro RP, 2018, Regeneration (Oxf), doi: 10.1002/reg2.98). Following the reviewer’s valuable suggestion, we edited the paragraphs of the introduction (line 38-39) and conclusion (line 265-270) for clarifying the definition of epimorphosis.

A more comprehensive rewriting of the manuscript is absolutely needed.

Response: A comprehensive rewriting of manuscript has been carried out and marked in red in the revised paper.

There are some important studies (see below) that can be used during the rewriting of the manuscript.

Response: Some of these were additionally cited in the paper.

Round 2

Reviewer 2 Report

Nice work, congratulation!

Only two mistakes occur in the text:

Line 190: This result suggests that organs located in the anterior and posterior                    segments grow actively.

Line 220: Nuclei are marked by DAPI staining (blue).